# Region of Attainable Redaction, an extension of Ellipse of Insignificance analysis for gauging impacts of data redaction in dichotomous outcome trials

**David Robert Grimes[1,2]***

[1]School of Medicine, Trinity College Dublin, Dublin, Ireland; [2]School of Physical Sciences, Dublin City University, Dublin, Ireland

**Abstract** In biomedical science, it is a reality that many published results do not withstand deeper investigation, and there is growing concern over a replicability crisis in science. Recently, Ellipse of Insignificance (EOI) analysis was introduced as a tool to allow researchers to gauge the robustness of reported results in dichotomous outcome design trials, giving precise deterministic values for the degree of miscoding between events and non-events tolerable simultaneously in both control and experimental arms (Grimes, 2022). While this is useful for situations where potential miscoding might transpire, it does not account for situations where apparently significant findings might result from accidental or deliberate data redaction in either the control or experimental arms of an experiment, or from missing data or systematic redaction. To address these scenarios, we introduce Region of Attainable Redaction (ROAR), a tool that extends EOI analysis to account for situations of potential data redaction. This produces a bounded cubic curve rather than an ellipse, and we outline how this can be used to identify potential redaction through an approach analogous to EOI. Applications are illustrated, and source code, including a web-based implementation that performs EOI and ROAR analysis in tandem for dichotomous outcome trials is provided.

**\*For correspondence:**
davidrobert.grimes@tcd.ie

**Competing interest:** The author declares that no competing interests exist.

## Editor's evaluation

This valuable study develops the Region of Attainable Redaction (ROAR), which quantifies the potential sensitivity of conclusions due to omitted data in two-arm clinical trials and studies of associations between dichotomous outcomes and exposures. The idea is supported by solid numerical examples and an application to a large meta-analysis. The concept of ROAR is a useful reminder of the fragility of some clinical findings.

## Introduction

Despite the crucial importance of biomedical science for human well-being, the uncomfortable reality is that swathes of published results in fields from psychology to cancer research are less robust than optimum (*Ioannidis, 2005*; *Krawczyk, 2015*; *Loken and Gelman, 2017*; *Grimes et al., 2018*; *Errington et al., 2021*). In cases when findings are spurious, inappropriate or errant statistical methods are often the primary cause of untrustworthy research, from incorrect interpretations of p-values to unsuitable tests to data redaction and under-reporting of overtesting, leading to research waste and unsound conclusions (*Hoffmann et al., 2013*; *Altman and Krzywinski, 2017*; *Colquhoun, 2014*; *Glasziou et al., 2014*; *Grimes and Heathers, 2021a*; *Itaya et al., 2022*; *Baer et al., 2021a*; *Baer et al., 2021b*). Across biomedical sciences, dichotomous outcome trials and studies are of paramount

importance, forming the basis of everything from preclinical observational studies to randomized controlled trials. Such investigations contrast experimental and control groups for a given intervention, comparing the numbers experiencing a particular event in both arms to infer whether differences between the intervention and control arm might exist.

Such investigations are vital, but concern has been raised over the fragility of many published works, where small amounts of recoding from event to non-event in experimental arms or vice versa in control arms can create an illusion of a relationship where none truly exists. In previous work by the author (***Grimes, 2022***), Ellipse of Insignificance (EOI) analysis was introduced as a refined fragility index capable of handling even huge data sets analytically with ease, considering both control and experimental arms simultaneously, which traditional fragility analysis cannot. EOI analysis is robust and analytical, suitable not only for Randomized Controlled Trial (RCT) analysis but for observational trials, cohort studies, and general preclinical work. Additionally, it also links the concept of fragility to test sensitivity and specificity when these are known for the detection of events, enabling investigators to probe not only whether a result is arbitrarily fragile, but to truly probe whether consider certain results are even possible. Accordingly, it yields both objective metrics for fragility and can be employed to detect inappropriate manipulation of results if the statistical properties of the tests used are known. A web implementation of this is available at https://drg85.shinyapps.io/EOIanalysis/, replete with code in *R* and other popular languages for general application.

While EOI analysis is a powerful method for ascertaining trial robustness, it does not explicitly consider the scenario where data is redacted. Data redaction in biomedical science creates spurious results and untrustworthy findings (***Grimes and Heathers, 2021b***), and can be difficult to detect. Data redaction can be accidental due to some systematic error in analysis, due to missing data, or arise through deliberate cherry-picking, and there are currently few tools for gauging its likely impact outside of direct simulation. In this technical note, we unveil a novel and powerful method for quantifying how much redaction would be required to explain a seemingly significant finding in dichotomous outcome trials, automatically finding the degree of redaction required to yield spurious results, and objective metrics for defining this. While EOI analysis produced a conic section in the form of an inclined ellipse where significance disappeared, this new tool instead produces a bounded region where significance disappears attainable by redaction and calculates the minimal vector to this regions. This technical note outlines the methodology of Region of Attainable Redaction (ROAR) analysis, including examples, R and MATLAB code for user implementations, and a web implementation for ease of deployability.

## Methods

The underlying geometrical and statistical basis for EOI analysis has been previously derived and described. In brief, EOI arises from chi-squared analysis, ascertaining how many participants in experimental and control groups could be recoded from events to non-events and vice versa before apparently significance was lost. This is a powerful approach for determining robustness of outcomes, and a web implementation and code are available at https://drg85.shinyapps.io/EOIanalysis/. EOI in its current form, however, does not consider the situation where a significant result might be obtained by data redaction, where an experimenter censors or neglects observations in the final analysis.

**Table 1.** Reported groups and related variables.

**Redaction for $RR_e > 1$**

|  | Endpoint positive | Endpoint negative |
| --- | --- | --- |
| Experimental group | $a$ | $b + x$ |
| Control group | $c + y$ | $d$ |

**Redaction for $RR_e < 1$**

|  | Endpoint positive | Endpoint negative |
| --- | --- | --- |
| Experimental group | $a + x$ | $b$ |
| Control group | $c$ | $d + y$ |

Defining $a$ as the reported data of endpoint positive cases in the experimental or exposure arm, $b$ as the reported experimental endpoint negative, $c$ as reported endpoint positive cases in the control arm, and $d$ as the reported endpoint negative cases in the control arm, we may define $x$ and $y$ as hypothetical redacted data in the experimental and control arm, respectively. We further define the total reported sample as $n = a + b + c + d$. To account for the impact of redaction, consider that an experimenter may obtain a significant result in favour of the experimental group in several ways. When relative risk is given by $\frac{a(c+d)}{c(a+b)}$ with no significant difference, the experimental arm could still yield a greater relative risk ($RR_E > 1$) than the control arm if either $x$ endpoint negative events had been jettisoned from the experimental arm, $y$ endpoint positive events jettisoned from the control or comparison arm, or a combination of both. Equally, if there is no significant difference but a lower relative risk in the experimental group is sought ($RR_E < 1$), such a finding can be manipulated by either jettisoning $x$ endpoint positive cases from the experimental arm, $y$ endpoint negative cases from the control arm, or a combination of both. These situations are given in *Table 1*. Risk ratio is used in this work for simplicity in gauging the relative impact of an ostensibly significant effect in the experimental arm and can be readily converted to odds ratio if preferred.

Applying the chi-square statistic outlined previously with a threshold critical value for significance of $\nu_c$, the resulting identity when $RR_E > 1$ is

$$\frac{(n + x + y)\Big(ad - (b + x)(c + y)\Big)^2}{(a + b + x)(c + d + y)(a + c + y)(b + d + x)} - \nu_c = 0 \tag{1}$$

and when $RR_E < 1$, the identity is

$$\frac{(n + x + y)\Big((a + x)(d + y) - bc\Big)^2}{(a + b + x)(c + d + y)(a + c + x)(b + d + y)} - \nu_c = 0 \tag{2}$$

Similar to EOI analysis, these forms can be expanded. However, the resultant equations in either case are not the conic sections of an inclined ellipse as with EOI analysis, but a more complicated cubic curve also in two variables. The resultant identity is $g(x, y)$, polynomial in $x$ and $y$, with the list of 15 coefficients in either case in given in the mathematical appendix (*Supplementary file 1*, Table S1). The region bound by this equation is the ROAR, and any $g(x, y) \leq 0$ changes an ostensibly significant finding to the null one, with $x$ and $y$ respectively yielding the redaction from the experimental and control group required.

## ROAR derivation and FOCK vector

In EOI analysis, we derived an analytical method for finding the minimum distance from the origin to the EOI. This point and vector, the Fewest Experimental/Control Knowingly Uncoded Participants (FECKUP) vector, allowed us to ascertain the minimal error which would render results insignificant. The resultant curve and bound region are inherently more complex in ROAR analysis, but the general principle remains. We seek the minimum distance from the origin to the region bound by $g(x, y) = 0$, defining the vector to this point $(x_e, y_e)$ as the Fewest Observations/Censored Knowledge (FOCK) vector. Unlike EOI analysis, we cannot exploit geometric arguments to solve this analytically, and instead we proceed by the method of Lagrange multipliers. The minimum distance from the origin to a point is given by

$$D(x, y) = \sqrt{x^2 + y^2} \tag{3}$$

Defining the polynomial defined in *Supplementary file 1*, Table S1 as $g(x, y)$, we can exploit the properties of Lagrange multipliers to write

$$\frac{\partial D}{\partial x} = \lambda \frac{\partial g}{\partial x} \tag{4}$$

$$\frac{\partial D}{\partial y} = \lambda \frac{\partial g}{\partial y}. \tag{5}$$

As we know $\lambda \neq 0$, we can rearrange these equations for the constant scalar $\lambda$ and equate them, subject to the constraint $g(x, y) = 0$. After rearrangement, we deduce that

$$y\frac{\partial g}{\partial x} - x\frac{\partial g}{\partial y} = 0. \tag{6}$$

This yields another unwieldy polynomial in two variables with 18 coefficients, listed in the mathematical appendix (*Supplementary file 1*, Table S2) for both cases. If we define the resultant function as $h(x, y)$, we seek to solve the simultaneous equations

$$g(x_e, y_e) = 0 \tag{7}$$

$$h(x_e, y_e) = 0. \tag{8}$$

While analytical solutions are likely intractable, this can be readily solved numerically subject to the constraints that $x_e > 0$ and $y_e > 0$. By Bézout's theorem, there are potentially up to 25 solutions to this simultaneous equation, so we restrict potential solution pairs to the real domain and select the pair yielding the minimum length FOCK vector, corresponding to $(x_e . y_e)$ as illustrated in *Figure 1*. Additionally, we solve $g(x_c, 0) = 0$ and $g(0, y_c) = 0$ as simple cubic equations to find the minimum number of observations redacted in exclusively the experimental and control groups to lose significance. The resolution of the FOCK vector yields the minimum redacted combination of experimental and control groups, given by

$$r_{min} = \lfloor x_e + y_e \rfloor. \tag{9}$$

## Metrics for degree of potential redaction

In EOI analysis, we established objective metrics to characterize the degree of potential miscoding required to sustain the null hypothesis. In this technical note, we establish analogous parameters. Considering only potential redaction in the experimental group, we define the degree of potential redaction that can be sustained while the null hypothesis remains rejected, given by

$$\rho_E = 1 - \frac{a+b}{a+b+x_c}. \tag{10}$$

For example, a ROAR analysis with $\rho_E = 0.1$ would inform us that at least 10% of experimental participants would have to be redacted for the result to lose significance. By similar reasoning, the tolerance threshold for error allowable in the control group is then

$$\rho_C = 1 - \frac{c+d}{c+d+y_c}. \tag{11}$$

Finally, errors in both the coding of the experimental and control group can be combined with FECKUP point knowledge. While $f_{min}$ gives a minimum vector distance to the ellipse, we instead take the length of the vector components to reflect to yield an absolute accuracy threshold of

$$\rho_A = 1 - \frac{n}{n + r_{min}}. \tag{12}$$

Unlike the EOI case, there is no direct relationship between test sensitivity/specificity and potential redaction.

## Application to large data sets and meta-analyses

ROAR is also highly effective with large data sets, and with certain caveats can be applied to even meta-analyses results. For a meta-analyses of $i$ dichotomous outcome trials, the crude pool risk ratio is given by

$$RR_C = \frac{\sum_1^i a_i \sum_1^i (c_i + d_i)}{\sum_1^i c_i \sum_1^i (a_i + b_i)} \tag{13}$$

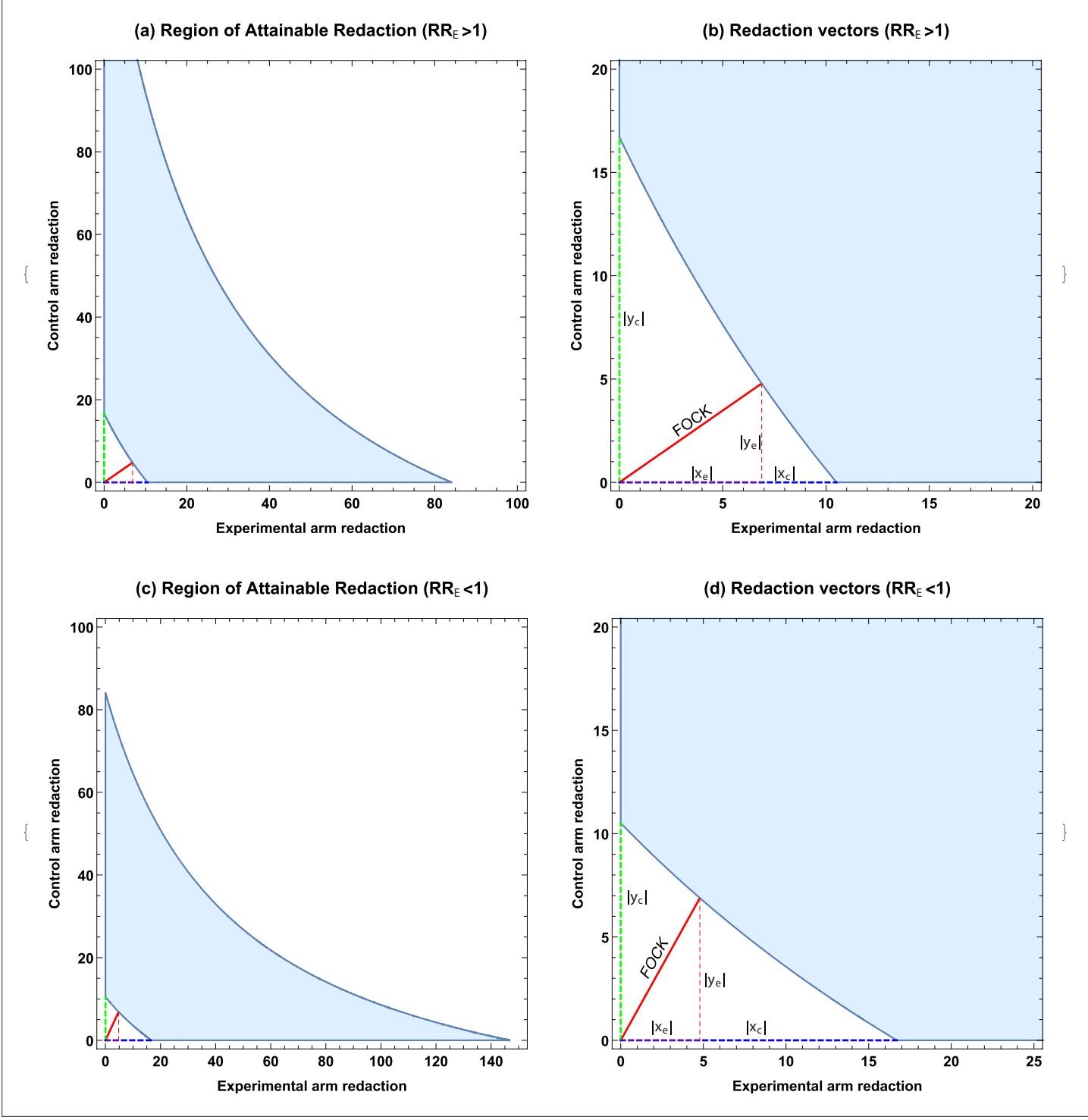

**Figure 1.** ROAR analysis implementation example. (**a**) A simulated example of the Region of Attainable Redaction (ROAR) for $a = 70$, $b = 30$, $c = 50$, $d = 50$ ($RR_E > 1$) with all points bounded by the shaded region depict a degree of redaction which would not lead to the null being rejected. (**b**) Relevant vectors for ascertaining possible redaction thresholds in $RR_E > 1$ case. (**c**) ROAR analysis of the similar data but with ($RR_E < 1$) ($a = 50$, $b = 50$, $c = 70$, $d = 30$). Note that $RR_E > 1$ case is a transform of $RR_E < 1$ situation. (**d**) Relevant vectors for ascertaining possible redaction thresholds in case $RR_e < 1$.

**Table 2.** ROAR-derived metrics for published data (see 'Results' for details).

| ROAR statistic ($\alpha = 0.05$) | Statistics for simulated example case | |
| --- | --- | --- |
| | Calculated ROAR value (simulated $RR_E > 1$ data) | Calculated ROAR value (simulated $RR_E < 1$ data) |
| Total subjects reported $N$ | 200 | 200 |
| Relative risk (95% CI) $RR_E$ | 1.40 (1.11–1.77) | 0.60 (0.46–0.76) |
| FOCK point | $(x_e, y_e) = (6.89, 4.79)$ | $(x_e, y_e) = (4.79, 6.89)$ |
| $r_{min}$ | 12 subjects | 12 subjects |
| Experimental redaction tolerance $\rho_E$ | 9.51% ($\lceil x_c \rceil$ = 10 subjects) | 14.32% ($\lceil x_c \rceil$ = 14 subjects) |
| Control redaction tolerance $\rho_C$ | 14.32% ($\lceil y_c \rceil$ = 16 subjects) | 9.51% ($\lceil y_c \rceil$ = 10 subjects) |
| Total redaction tolerance $\rho_A$ | 5.66% (12 subjects) | 5.66% (12 subjects) |

| ROAR statistic ($\alpha = 0.05$) | Statistics for large meta-analysis |
| --- | --- |
| | Calculated ROAR value |
| Total subjects reported $N$ | 39,197 |
| Relative risk (95% CI) | 0.85 (0.74–0.97) |
| FOCK point | $(x_e, y_e) = (14.17, 0.32)$ |
| $r_{min}$ | 14 subjects |
| Experimental redaction tolerance $\rho_E$ | 0.07% (13 subjects) |
| Control redaction tolerance $\rho_C$ | 3.21% (649 subjects) |
| Total redaction tolerance $\rho_A$ | 0.04% (14 subjects) |

FOCK, Fewest Observations/Censored Knowledge; ROAR, Region of Attainable Redaction.

whereas the Cochran–Mantel–Haenszel adjusted risk ratio accounts for potential confounding between studies and adjusts for sample size, given by

$$RR_{CMH} = \frac{\sum_1^i \frac{a_i(c_i + d_i)}{n_i}}{\sum_1^i \frac{c_i(a_i + b_i)}{n_i}}. \tag{14}$$

The magnitude of confounding between studies is given by $|1 - \frac{RR_{CMH}}{RR_C}|$. If this is small (typically <10%), confounding can be assumed minimal and the crude ratio used, allowing ROAR to be deployed directly on pooled meta-analyses results if these conditions are met. When confounding is significant between studies, direct ROAR is not applicable and these caveats are expanded upon in discussion.

## Results

### Example deployment and ROAR behaviour

To demonstrate the usage of ROAR, we consider a simple twin example with the following arrangement of data.

1. $RR_E > 1$: We generate a data set of $N = 200$, with $a = 70$, $b = 30$ in the experimental arm, and $c = d = 50$ in the control arm. This yields p<0.004, and a hypothetical risk ratio of 1.4 (95% confidence interval: 1.11–1.77). The ROAR for this data set is illustrated in *Figure 1a and b*, with the degree of redaction required given in *Table 2*s.
2. $RR_E < 1$: We generate a similar data set of $N = 200$, but invert the experimental and control arm so that $a = b = 50$ with $c = 70$ and $d = 30$ in the control arm. This also yields p<0.039, and a hypothetical risk ratio of 0.71 (95% confidence interval: 0.57–0.90). The ROAR for this data set is illustrated in *Figure 1c and d*, with the degree of redaction required given in *Table 2*.

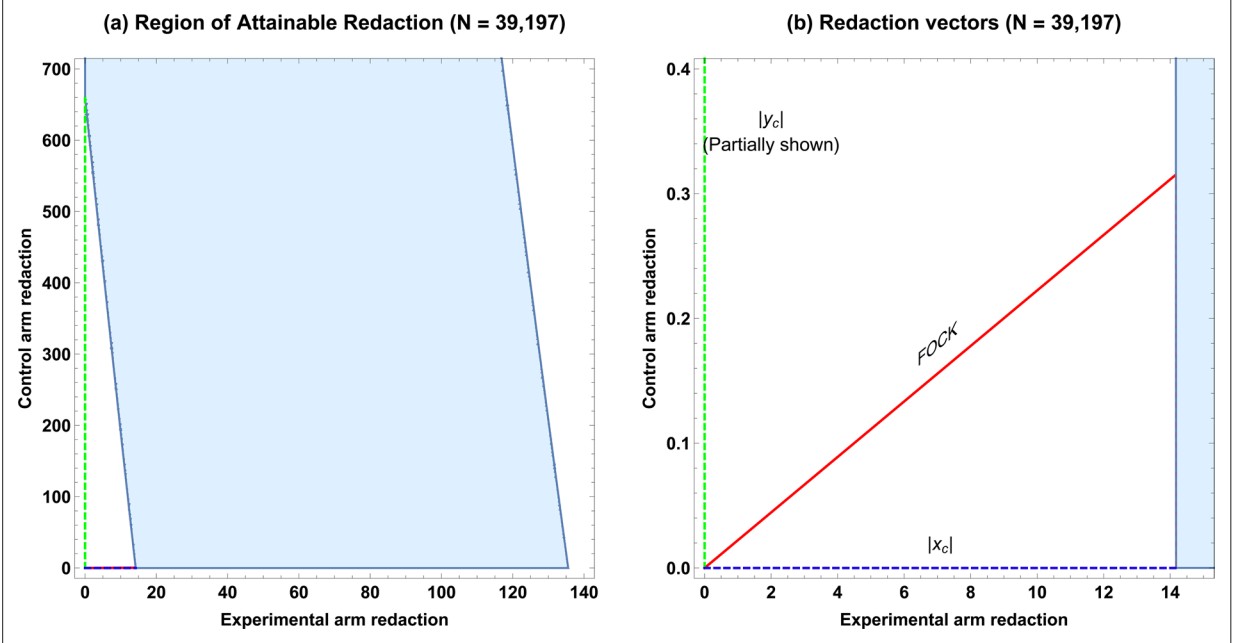

**Figure 2.** ROAR example for large data-set. (**a**) A simulated example of the Region of Attainable Redaction (ROAR) for a meta-analysis of $N = 39,197$ as described in the text. (**b**) Relevant vectors for ascertaining possible redaction thresholds.

As can be seen from **Figure 1**, the cases $RR_E > 1$ and $RR_E < 1$ are essentially geometrical rotations and reflections of one another, with $r_{min}$ the same in both the values of $\rho_E$ and $\rho_C$ being transposed on reflection, as it evident from **Table 2**. This showcases the general behaviour of ROAR analysis, and in this example, it would require a redaction of between 10 and 16 subjects to lose apparent significance in either case, requiring at least 5.66% of the total subjects to have been redacted. Note that the real values of $x_e$ and $y_e$ are employed in calculating $\rho_E$ and $\rho_C$, whereas the integer value $r_{min}$ is used in $\rho_A$. For $x_c$ and $y_c$, integer ceiling values yield the greatest possible redaction.

## Application to large data sets and meta-analyses

We consider a published meta-analysis of vitamin D supplementation on cancer mortality (**Zhang et al., 2019**), comprising of $N = 39,197$ patients from five RCTs, with $a = 397$ deaths in the experimental group (supplementation) versus $b = 19,204$ non-deaths and $c = 468$ deaths in the control group versus $d = 19,128$ non-deaths. Although the authors did not see reduction in all-cause mortality, subanalysis for the cancer population yielded an odds ratio of 0.85 (95% confidence interval: 0.74–0.97 for supplementation), reporting an $\approx$ 15% reduction in cancer mortality risk. With $RR_C = 0.8481$, $RR_{CMH} = 0.8474$, the magnitude of confounding is < 0.09% and thus we can apply ROAR to the pooled data. In this case, ROAR is illustrated in **Figure 2**, and the degree of redaction required is given in **Table 2**. Despite the ostensible strength of the result and large sample size, redaction of a mere 14 subjects (0.036%) or a small fraction of missing data would be sufficient to nullify the apparent finding, despite it stemming from a large meta-analysis.

## Browser-based implementation and source code

A browser-based implementation combining both EOI and ROAR is hosted at https://drg85.shin-yapps.io/EOIROAR/, and relevant source code is hosted online for languages including *R*, MATLAB/OCTAVE, and Mathematica at https://github.com/drg85/EOIROAR_code, copy archieved at **Grimes, 2023**.

## Discussion

ROAR analysis outlined in this technical note extends the functionality and usefulness of EOI analysis, allowing users to estimate the likely impacts of missing data. EOI analysis and related fragility

methods had the limitations that while they handled potential miscoding, they were unsuitable for inferences or quantification of impacts of redacted data or subjects lost to follow-up. Accordingly, ROAR is a powerful method of gauging the potential impacts of missing data. While more mathematically complex than EOI analysis, ROAR remains deterministic and rapid, but shares the limitation that it is only currently applicable to dichotomous outcome trials and studies, and should be applied very cautiously to time-to-event data, where it may not be suitable. Like EOI analysis, ROAR differs from typical fragility metrics by avoiding Fisher's exact test as this is not suitable for large data sets which EOI and ROAR can readily handle. This is typically not a problem as the chi-squared test employed approximates Fisher's test in most circumstances. However, like EOI analysis, p-values for small trials can differ slightly from chi-squared result. ROAR analysis is built upon chi-squared statistics, and it is thus possible for edge cases of small numbers to yield discordant results with Fisher's exact test also. This can be shown from a theoretical standpoint to not typically make any appreciable difference except for rare events in very small trials (*Grimes, 2022*; *Baer et al., 2021a*).

Application of ROAR analysis is inherently context specific. For clinical trials, preregistration in principle reduces the potential for experimenter choices like redaction bias changing the outcome. But the implementation of preregistered protocols does not exclude the possibility of data dredging in the form of p-hacking or HARKing (hypothesizing after the results are known). Researchers rarely follow the precise methods, plan, and analyses that they preregistered. A recent analysis found that pre-registered studies, despite having power analysis and higher sample size than non-registered studies, do not *a priori* seem to prevent p-hacking and HARKing (*Bakker et al., 2020*; *Singh et al., 2021*; *El-Boghdadly et al., 2018*; *Sun et al., 2019*), with similar proportions of positive results and effect sizes between preregistered and non-preregistered studies (*van den Akker et al., 2023*). A survey of 27 preregistered studies found researchers deviating from preregistered plans in all cases, most frequently in relation to planned sample size and exclusion criteria (*Claesen et al., 2021*). This latter aspect lends itself to potential redaction bias (*Grimes and Heathers, 2021b*), which can be systematic rather than deliberate and thus a means to quantify its impact is important. More importantly, EOI analysis has application for dichotomous beyond clinical trials. In preclinical work, cohort, and observational studies, the scope for redaction bias greatly increases as reporting and selection of data falls entirely on the experimenter, and thus methods like ROAR to probe potential redaction bias are important. ROAR also has potential application in case–control studies, where selection of an inappropriately fragile control group could give spurious conclusions. This again comes with caveats as studies adjusted for potential confounders and predictors might make ROAR inappropriate in such cases.

As demonstrated in this work, ROAR is under certain circumstances applicable even to large meta-analysis. In this instance, a potential redaction of just 14 subjects from over 39,000 was sufficient to overturn the null hypothesis, despite a relative risk reduction of 15% being widely reported on the basis of this meta-analysis. This of course is not to say that any redaction occurred only to quantify the vulnerability of such a study to missing data. While beyond the scope of this technical note, it is worth noting that a subsequent 2022 meta-analysis (*Zhang et al., 2022*) of 11 RCTS (including those in the 2019 [*Zhang et al., 2019*] meta-analysis) found no significant reduction in cancer mortality with vitamin D supplementation, a potential testament to the need to consider the fragility of results to missing or miscoded data, even with ostensibly large samples. There are however important caveats to applying ROAR to meta-analyses. In its naive form, it is only suitable when there is minimal confounding between studies so that the crude relative risk differs minimally from the adjust risk ($RR_C \approx RR_{CMH}$), such as in the illustrative work considered herein. When this is not the case, results from individual studies cannot be crudely pooled and ROAR is not valid in these instances. As ROAR applied to meta-analyses pools studies into a simple crude measure, it does not identity the particular study or studies where hypothetical redaction might have occurred, only the global fragility metric. A full theoretical extension of ROAR and EOI for specifically for meta-analyses is beyond the scope of this work, and accordingly, ROAR must be cautiously implemented and carefully interpreted in investigations of meta-analyses.

As discussed in the EOI paper (*Grimes, 2022*), poor research conduct including inappropriate statistical manipulation and data redaction are not uncommon, affecting up to three quarters of all biomedical science (*Fanelli, 2009*). Like EOI analysis, the ROAR extension has a potential role in detecting manipulations that nudge results towards significance, and identifying inconsistencies in

data and adding invaluable context. It is a demonstrable reality that even seemingly strong results can falter under inspection, and tools like ROAR and EOI analysis have a potentially important role in identifying weak results and statistical inconsistencies, with wide potential application across meta-research with the goal of furthering sustainable, reproducible research.

## Acknowledgements

DRG thanks the Wellcome trust for their support, and Profs Phillip Boonstra and Detlef Weigel for their helpful comments.

## Additional information

### Funding

| Funder | Grant reference number | Author |
| --- | --- | --- |
| Wellcome Trust | 214461/A/18/Z | David Robert Grimes |

The funders had no role in study design, data collection and interpretation, or the decision to submit the work for publication. For the purpose of Open Access, the authors have applied a CC BY public copyright license to any Author Accepted Manuscript version arising from this submission.

### Author contributions

David Robert Grimes, Conceptualization, Resources, Software, Formal analysis, Funding acquisition, Validation, Investigation, Visualization, Methodology, Writing – original draft, Project administration

### Author ORCIDs

David Robert Grimes ⓘ http://orcid.org/0000-0003-3140-3278

### Decision letter and Author response

Decision letter https://doi.org/10.7554/eLife.93050.sa1
Author response https://doi.org/10.7554/eLife.93050.sa2

## Additional files

### Supplementary files

- MDAR checklist
- Supplementary file 1. Coefficients for $g(x, y)$ (Table S1) and related identity $h(x, y)$ (Table S2).

### Data availability

Sample R / MATLAB / OCTAVE code and functions for rapid implementation of EOI analysis method outlined, hosted online at https://github.com/drg85/EOIROAR_code (copy archived at *Grimes, 2023*). Web implementation is available at: https://drg85.shinyapps.io/EOIROAR/. All data is available in the paper and on GitHub.

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
