## [Editor Report]

This valuable study develops the Region of Attainable Redaction (ROAR), which quantifies the potential sensitivity of conclusions due to omitted data in two-arm clinical trials and studies of associations between dichotomous outcomes and exposures. The idea is supported by solid numerical examples and an application to a large meta-analysis. The concept of ROAR is a useful reminder of the fragility of some clinical findings.

---

## [Decision Letter]

**Decision letter after peer review:**

Thank you for submitting your article "Region of Attainable Redaction, an extension of Ellipse of Insignificance analysis for gauging impacts of data redaction in dichotomous outcome trials" for consideration by *eLife*. I apologize for the long delay in reviewing your article. I had a great deal of trouble identifying reviewers who were willing and available to review. I (Phil) have reviewed your article as the Reviewing Editor, and the evaluation has been overseen Detlef Weigel as the Senior Editor.

Please see my recommendations below to help you prepare a revised submission.

*Reviewer #1 (Recommendations for the authors):*

In this article, Dr. Grimes develops the Region of Attainable Redaction (ROAR), which is a simple metric for the potential sensitivity of reported findings in two-arm randomized clinical trials due to omitted data. The idea of ROAR is to quantify the minimum number of hypothetical observations that – had they existed but not been reported – would have changed a statistically significant finding to a non-statistically significant finding. The author helpfully includes an easy-to-use online app and, for those interested, computer code. The strengths of this work include its ease of use and its intuitive meaning. Weaknesses include its limited scope of application, as discussed below.

The method is nominally applicable to any analysis of a binary outcome with two exposures in which a chi-squared test is appropriate. The intended use is for randomized clinical trials, but the utility of ROAR seems limited here given the expected rigor and reporting requirements for such studies, including preregistration, multiple levels of scientific and ethical review, and reporting requirements. Given this, it seems unlikely that the published analysis of a clinical trial could successfully "hide" observations. Although the Discussion notes that it is applicable to cohort and ecological studies, presumably any such observational study would adjust for other predictors and potential confounders in addition to the exposure of interest. Thus, it is not clear how applicable the concept of ROAR, which is an unadjusted analysis, is for such studies.

Dr. Grimes also applies ROAR to a large meta-analysis of vitamin D supplementation (exposure) and cancer mortality (outcome). This application seems more appropriate than those mentioned above. The key finding is that, in a meta-analysis of more than 39000 patients with 865 deaths, the statistical significance of the of 0.85 estimated risk ratio would be lost had there been just 14 additional (unreported) subjects in the exposure arm who had the outcome. Driving this surprising result is the low baseline risk of mortality in either group. For more prevalent events, the number would be much greater than 14.

A final weakness is in the author's claim in the Discussion that "[ROAR] can be employed…to detect the fingerprints of questionable research practices and even research fraud." This claim is surprising and not substantiated in the article, as ROAR is presented as a sensitivity analysis and not a diagnostic tool.

– Is there utility for single trials which require rigorous definitions and reporting of enrolled, eligible, evaluable patients? It seems unlikely that subjects could simply be redacted from a particular trial.

– Why do the regions in Figures 1 and 2 have upper limits? Does this correspond to a switching of significance in the other direction?

– Tables 2 and 3 are probably not of interest to most *eLife* readers and could probably be moved to a supplement.

– The notation should be made clearer: a, b, c, and d are the reported data, whereas x and y are the hypothetical redacted data. I don't think this is explicitly stated anywhere. Also, I don't think n is defined anywhere, presumably, it is a+b+c+d.

– In the second sentence of the second paragraph of the methods, I think the grammar needs to be clarified to reflect the hypothetical nature of the statement. For example, I think instead of '…are jettisoned…' the past perfect form '…had been jettisoned…' would be more appropriate. But more than just this needs to be changed.

– In section "ROAR derivation and FOCK vector", fifth line down: the e in 'ye' should be subscripted.

– In section "Results", in points 1 and 2: it is written "p < 0039" presumably there is missing a decimal somewhere. And this should be an equality not an inequality.

– In section Introduction, the sentence "EOI analysis in robust and analytical, suitable not only for RCT analysis but for observational trials, cohort studies, and general preclinical work". Should be changed to "…is robust and analytical".

– In Table 4, the percentages are not clear to me. E.g. Experimental redaction tolerance 9.51%(10 subjects). I thought the tolerance would be 10/110 or 9.1%.

– I think I had a similar comment on the EOI paper, but the FOCK coordinates seem like they should be integer-valued in order to be clinically applicable since you cannot redact a fraction of an observation. Put differently, I think that the identification of x_e and y_e and r_min should comprise three steps: (i) first identify x_e* and y_e*, the real-valued solutions to g(x,y)=0 and then (ii) identify (x_e, y_e) as the integer-valued coordinates that are closest to x_e* and y_e* and still in the ROAR. Then (iii) r_min = x_e + y_e. For example, in the bottom half of table 4, you cannot redact a fraction of an observation,

– Can ROAR be used in case-control studies – if so what special considerations would there need to be? And what about other estimands besides a risk ratio? Can you add some discussion about this?

---

## [Author Response]

Reviewer #1 (Recommendations for the authors):In this article, Dr. Grimes develops the Region of Attainable Redaction (ROAR), which is a simple metric for the potential sensitivity of reported findings in two-arm randomized clinical trials due to omitted data. The idea of ROAR is to quantify the minimum number of hypothetical observations that – had they existed but not been reported – would have changed a statistically significant finding to a non-statistically significant finding. The author helpfully includes an easy-to-use online app and, for those interested, computer code. The strengths of this work include its ease of use and its intuitive meaning. Weaknesses include its limited scope of application, as discussed below.

This is a concise and accurate summary, I thank the reviewer for it and will try to address their queries in order in this document.

The method is nominally applicable to any analysis of a binary outcome with two exposures in which a chi-squared test is appropriate. The intended use is for randomized clinical trials, but the utility of ROAR seems limited here given the expected rigor and reporting requirements for such studies, including preregistration, multiple levels of scientific and ethical review, and reporting requirements. Given this, it seems unlikely that the published analysis of a clinical trial could successfully "hide" observations. Although the Discussion notes that it is applicable to cohort and ecological studies, presumably any such observational study would adjust for other predictors and potential confounders in addition to the exposure of interest. Thus, it is not clear how applicable the concept of ROAR, which is an unadjusted analysis, is for such studies.

It is true that preregistration of clinical trials should, in principle, stem the influence of an experimenter choice changing the outcome. But there are caveats to this. Implementation of preregistered protocols does not exclude the possibility of p-hacking. Researchers rarely follow the precise methods, plan, and analyses that they preregistered. A recent analysis found that pre-registered studies, despite having power analysis and higher sample size than do not a prior seem to prevent p-hacking and HARKing, with similar proportions of positive results and effect sizes between preregistered and non-preregistered studies A 2019 survey of 27 preregistered studies found researchers deviating from preregistered plans in all cases, most frequently in relation to planned sample size and exclusion criteria. This latter aspect lends itself to potential redaction bias, which can be systematic rather than deliberate and thus a means to quantify its impact is important. In preclinical work and cohort studies, the scope for redaction bias greatly increases as reporting and selection of data falls entirely on the experimenter. The point about observational and ecological studies is absolutely valid, and it would depend on the design whether ROAR could be applied. Accordingly, the discussion has been revised with the following additional text to address this:

“Application of ROAR analysis is inherently context specific. For clinical trials, preregistration in principle reduces the potential for experimenter choices like redaction bias changing the outcome. But the implementation of preregistered protocols does not exclude the possibility of data dredging in the form of p-hacking or HARKing (hypothesizing after the results are known). Researchers rarely follow the precise methods, plan, and analyses that they preregistered. A recent analysis found that pre-registered studies, despite having power analysis and higher sample size than do not a prior seem to prevent p-hacking and HARKing^17-20^, with similar proportions of positive results and effect sizes between preregistered and non-preregistered studies^21^ A survey of 27 preregistered studies found researchers deviating from preregistered plans in all cases, most frequently in relation to planned sample size and exclusion criteria^22^ This latter aspect lends itself to potential redaction bias^15^ which can be systematic rather than deliberate and thus a means to quantify its impact is important. More importantly, EOI analysis has application for dichotomous beyond clinical trials. In preclinical work, cohort, and observational studies, the scope for redaction bias greatly increases as reporting and selection of data falls entirely on the experimenter, and thus methods like ROAR to probe potential redaction bias are important. ROAR also has potential application in case control studies, where selection of an inappropriately fragile control group could give spurious conclusions. This again comes with caveats, as studies adjusted for potential confounders and predictors might make ROAR inappropriate in such cases.”

Dr. Grimes also applies ROAR to a large meta-analysis of vitamin D supplementation (exposure) and cancer mortality (outcome). This application seems more appropriate than those mentioned above. The key finding is that, in a meta-analysis of more than 39000 patients with 865 deaths, the statistical significance of the of 0.85 estimated risk ratio would be lost had there been just 14 additional (unreported) subjects in the exposure arm who had the outcome. Driving this surprising result is the low baseline risk of mortality in either group. For more prevalent events, the number would be much greater than 14.

This is true, but the converse is that where events less rare, such an analysis would suggest the results to be robust. Please note that this discussion has changed substantially in this revision, because upon reflection, it pivots on the assumption there is little confounding between the studies that make up the meta-analyses. To reflect this, the text has changed in the methodology, results, and Discussion sections.

A final weakness is in the author's claim in the Discussion that "[ROAR] can be employed…to detect the fingerprints of questionable research practices and even research fraud." This claim is surprising and not substantiated in the article, as ROAR is presented as a sensitivity analysis and not a diagnostic tool.

This is a justified criticism, and it was badly phrased, apologies. The idea was that it might added to the arsenal of teams like INSPECT-SR in their efforts to detect dubious trials, but without context what I had wrote overstated things. I’ve deleted the offending sentence as I don’t think it adds anything and it is ripe for causing confusion, apologies.

– Is there utility for single trials which require rigorous definitions and reporting of enrolled, eligible, evaluable patients? It seems unlikely that subjects could simply be redacted from a particular trial.

As mentioned in the new references 17-22 inclusive, even with preregistration it is possible to p-hack results, and the redaction of prespecified data is one of the most common reasons for this. Even if not intention, as discussed in reference 15, systematic experimenter choices in excluding data can lend themselves to this (reference 15). To better address this, I’ve modified the text as outlined in my response to point 2.

– Why do the regions in Figures 1 and 2 have upper limits? Does this correspond to a switching of significance in the other direction?

Yes this is right, if I’m understanding the question correctly. In an analogous way to EOI, if one redacts enough they go from losing significance to “flipping” it to highly significant in the other direction the equivalent of going from a fractional relative risk not encompassing unity to a relative risk greater than one also not encompassing unity. For example, in figure 1a, redacting 16 people from the experimental arm would lose significance, whereas redacting 150 would be significant but in the opposite direction from the original reported. The FOCK vector finds the minimal distance to the region of redaction, the rmin resolves this, as explored further in point 14.

– Tables 2 and 3 are probably not of interest to most eLife readers and could probably be moved to a supplement.

They have now been moved to a mathematical appendix as they were rather space consuming.

– The notation should be made clearer: a, b, c, and d are the reported data, whereas x and y are the hypothetical redacted data. I don't think this is explicitly stated anywhere. Also, I don't think n is defined anywhere, presumably, it is a+b+c+d.

Thank you for noticing this, a total oversight on my behalf. The text has now been changed to read:

“Defining a as the reported data of endpoint positive cases in the experimental or exposure arm, b as the reported experimental endpoint negative, c as reported endpoint positive cases in the control arm and d as the reported endpoint negative cases in the control arm, we may define x and y as hypothetical redacted data in the experimental and control arm respectively. We further define the total reported sample as n = a + b + c + d.”

– In the second sentence of the second paragraph of the methods, I think the grammar needs to be clarified to reflect the hypothetical nature of the statement. For example, I think instead of '…are jettisoned…' the past perfect form '…had been jettisoned…' would be more appropriate. But more than just this needs to be changed.

Yes, this sounds much better, thank you. It has been changed throughout the text for consistency.

– In section "ROAR derivation and FOCK vector", fifth line down: the e in 'ye' should be subscripted.

Well spotted! Completely missed this, corrected now.

– In section "Results", in points 1 and 2: it is written "p < 0039" presumably there is missing a decimal somewhere. And this should be an equality not an inequality.

Thank you, it is actually something in the region of p = 0.00389 so I have rewritten this as p < 0.004 for clarity in this iteration.

– In section Introduction, the sentence "EOI analysis in robust and analytical, suitable not only for RCT analysis but for observational trials, cohort studies, and general preclinical work". Should be changed to "…is robust and analytical".

Corrected, thank you for spotting this.

– In Table 4, the percentages are not clear to me. E.g. Experimental redaction tolerance 9.51%(10 subjects). I thought the tolerance would be 10/110 or 9.1%.

The slight discrepancy arises because ρE and ρC are calculated with the real-valued intersection point (xe,ye) whereas ρA arises from the integer valued rmin. This has now been clarified in the text, and is expanded upon in point 14 below.

– I think I had a similar comment on the EOI paper, but the FOCK coordinates seem like they should be integer-valued in order to be clinically applicable since you cannot redact a fraction of an observation. Put differently, I think that the identification of x_e and y_e and r_min should comprise three steps: (i) first identify x_e* and y_e*, the real-valued solutions to g(x,y)=0 and then (ii) identify (x_e, y_e) as the integer-valued coordinates that are closest to x_e* and y_e* and still in the ROAR. Then (iii) r_min = x_e + y_e. For example, in the bottom half of table 4, you cannot redact a fraction of an observation,

This is a fair point, but the current terminology is used to keep it consistent with EOI. While (xe,ye) is real valued, its resolved vector is the integer valued rmin is analogy with the resolved FECKUP vector in EOI. The reason it is only resolved after this step is because there will be edge cases where taking the floor or ceiling values (xe,ye) before the vector resolution will offset the resolved vector by one-two subjects. But to clarify that (xc,yc) must be integer valued, this has been added to the table and text to avoid confusion.

– Can ROAR be used in case-control studies – if so what special considerations would there need to be? And what about other estimands besides a risk ratio? Can you add some discussion about this?

Please see response to point 2 in relation to case-control studies. Risk ratio was employed as it gives an intuitive metric for experimental interventions that have an apparently significant effect, but one could readily reformulate the entire thing as an odds ratio or similar. To account for this, I added the following line to the methodology:

“Risk ratio is used in this work for simplicity in gauging the relative impact of an ostensibly significant effect in the experimental arm, and can be readily be converted to odds ratio if preferred.”